# Growth Assessment in Preterm Children from Birth to Preschool Age

**DOI:** 10.3390/nu12071941

**Published:** 2020-06-30

**Authors:** Simone Ceratto, Francesco Savino, Silvia Vannelli, Luisa De Sanctis, Francesca Giuliani

**Affiliations:** 1Postgraduate School of Pediatrics, University of Torino, 10126 Turin, Italy; simone.ceratto@gmail.com; 2Division of Pediatrics and Neonatology, Department of Maternal Medicine, Nuovo Ospedale degli Infermi, 13875 Ponderano (Biella), Italy; 3Early Infancy Special Care Unit, Regina Margherita Children Hospital, A.O.U. Città della Salute e della Scienza di Torino, 10126 Torino, Italy; francesco.savino@unito.it; 4Pediatric Endocrinology and Diabetology Unit, Regina Margherita Children Hospital, AOU Città della Salute e della Scienza di Torino, 10126 Torino, Italy; svannelli@cittadellasalute.to.it (S.V.); luisa.desanctis@unito.it (L.D.S.); 5Pediatric Endocrinology, Department of Public Health and Pediatric Sciences, University of Torino, 10124 Torino, Italy

**Keywords:** preterm infants, postnatal growth, extrauterine growth restriction, INTERGROWTH-21st, growth curves, SGA

## Abstract

Preterm infant growth is a major health indicator and needs to be monitored with an appropriate growth curve to achieve the best developmental and growth potential while avoiding excessive caloric intake that is linked to metabolic syndrome and hypertension later in life. New international standards for size at birth and postnatal growth for preterm infants are available and need implementation in clinical practice. A prospective, single center observational study was conducted to evaluate the in-hospital and long-term growth of 80 preterm infants with a mean gestational age of 33.3 ± 2.2 weeks, 57% males. Size at birth and at discharge were assessed using the INTERGROWTH-21^ST^ standards, at preschool age with World Health Organization (WHO) child growth standards. The employment of INTERGROWTH-21^ST^ Preterm Postnatal longitudinal standards during the in-hospital follow-up significantly reduced the diagnosis of short term extrauterine growth restriction when compared to commonly used cross sectional neonatal charts, with significant lower loss of percentiles between birth and term corrected age (*p* < 0.0001). The implementation of a package of standards at birth, preterm postnatal growth standards and WHO child growth standards proved to be consistent, with correlation between centile at birth and at follow-up, and therefore effective in monitoring growth in a moderate and late preterm infant cohort without chronic or major morbidities. Infants identified as small for gestational age at birth showed significantly more frequently a need for auxological referral.

## 1. Background

Monitoring preterm infant growth is a key point in the assessment of these children during hospitalization and after discharge from neonatal intensive care unit. [1,2]

Weight gain evaluation is used in clinical practice as a guidance to individualize nutritional intakes: as a consequence, the choice of an appropriate reference growth curve is key to achieve the best developmental and growth potential while avoiding excessive caloric intake, that is linked to metabolic syndrome and hypertension later in life. [3,4]

In the past, fetal growth was considered the standard of growth also for the preterm neonate; [5] nowadays, it is believed that such a target of weight gain in the first weeks of life may have negative consequences and “program” the infant to later increased incidence of overweight and cardiovascular risk. [6]

International standards for size at birth and postnatal growth standards for preterm infants have been developed within the INTERGROWTH-21^ST^ Project [7], using the same methodological approach of the World Health Organization (WHO) child growth standards, and are now available for use in clinical practice. The standards have been constructed with multiethnic population of infants without major complications and born to healthy, well-educated mothers without pregnancy complications known to affect fetal growth. The obtained standards, specific for preterms, are the most logical charts to be used for preterm infants and are meant to provide a growth reference that is realistic and should lead to less diagnoses of the so-called “extra-uterine growth restriction”, by definition based on the concept of using fetal growth as a standard for the preterm infants. [6,8] Testing the latest statement was therefore one of the aims of the present study, hypothesizing that extrauterine growth restriction (EUGR) is not strongly predictive of adverse later growth outcome if it is calculated using inappropriate charts such as neonatal size at birth charts, and that moderate and late preterm without short-term EUGR based on specific preterm postnatal charts will show adequate growth achievements at preschool age.

Finally, since we are dealing with relatively new size at birth standards [9,10], there is a need to validate the 10th percentile cutoff to define small for gestational age (SGA) and confirm whether it predicts later growth outcome.

To summarize, the present observational study was conducted in order to (1) evaluate the in-hospital growth of a population of preterm infants and the prevalence of growth failure at discharge using preterm standards compared to widely used size-at-birth charts [11,12]; (2) assess the long-term growth of such infants and the predictivity of INTERGROWTH-21^ST^ centiles at birth [9,10] of the centile reached in preschool age on the WHO standards [13,14]; (3) verify if being born small for gestational age on the new standards predicts an adverse outcome in a long-term follow-up.

## 2. Materials and Methods

All infants admitted after birth to the Special Care Unit for Infants of Regina Margherita Children Hospital, Turin, Italy who have reached an age of 4 to 5 years between 1st December 2018 and 31st December 2019 (3 months deviation from the indicated ages was tolerated) were assessed for inclusion in this prospective, single center observational study.

Exclusion criteria were:Gestational age at birth ≥ 37+0 weeks;Syndromes, conditions or diseases known to affect growth and/or requiring specific charts, e.g., Down’s syndrome or other genetic syndromes, short bowel syndrome, etc.;Chronic diseases which required prolonged parenteral nutrition.

A total of 237 children were assessed for eligibility. A total of 124 subjects were preterm and met inclusion criteria and parents were called by phone in order to invite them to take part into the study: of those, 33 had invalid phone number, 11 refused to participate. Finally, 80 infants born preterm were included in the study.

Obstetric antenatal charts were reviewed in order to record gestation complications and IUGR, defined according to the recommendations of the American College of Obstetricians and Gynecologists [15]

Neonatal records were also examined to get exact data about length of hospital stay, feeding practices during hospitalization and early morbidities.

Weight, length or height and head circumference were taken at birth as well as at term corrected age and at the follow-up visit by a trained anthropometrist using an electronic scale, Harpenden stadiometer and SECA craniometer (non-extensible tape). Parents’ heights were measured in order to calculate child’s statural target [13,14].

At the follow-up visit all participants underwent a physical examination. Parents completed a questionnaire enquiring data on the child’s health, feeding practices, acquired vaccinations and reaching the developmental milestones.

Anthropometry at birth was plotted on INTERGROWTH-21st size at birth standards [9,10], anthropometry 40 weeks postmenstrual age (± 2 weeks) was evaluated using Preterm Postnatal INTERGROWTH-21st standards [11] and Fenton charts [12] for comparison. Measures at follow-up visits were plotted on WHO child growth standards. [13,14] One follow-up visit was originally planned at preschool age, but if the infant showed unsatisfactory growth, as defined below, another visit was scheduled in order to calculate growth velocity.

Children with unsatisfactory growth were referred to auxological follow-up with the following criteria: statural height lower than third percentile on WHO standards and/or height-growth-percentile falling outside parent targets and/or reduced growth velocity for sex and age calculated longitudinally with at least two measurements during follow-up. [13,14] Parent targets were calculated with the Tanner equation [16].

Small for gestational age (SGA) was defined as with birth weight below the 10th percentile [17,18].

Parents or guardians gave informed consent for inclusion of their children before they participated in the study. The study was conducted in accordance with the Declaration of Helsinki and the protocol was approved by the Ethics Committee of Torino, code 219/2010.

Depending on the variable and on its distribution, *t*-test, chi-squared test or analysis of variance (ANOVA), for example to assess the correlation between measures at birth and at follow-up. Statistical analysis was conducted using IBM^®^ SPSS^®^ Statistics 24.0 software.

## 3. Results

### 3.1. Population Characteristics and in-Hospital Growth

Eighty children (57.5% male and 42.5% female) were recruited. Average age at the time of clinical evaluation was 4.21 ± 0.28 years. Additional baseline characteristics are reported in Table 1.

In order to assess the so-called extrauterine growth restriction, the variation of weight percentiles between birth and term “corrected” age was calculated, comparing results obtained using INTERGROWTH-21st [9,10,11] and Fenton’s charts [12]. According to INTERGROWTH-21st average loss was 10.25 ± 23.99 percentiles, while according to Fenton charts it was 19.34 ± 23.20 (*p* < 0.0001).

### 3.2. Follow-Up at Preschool Age

Percentiles of weight, length/height and head circumference at birth and at recruitment (approximately at 4 years of age) were evaluated, resulting normally distributed. Percentiles at birth (according to INTERGROWTH-21st) [9,10] and at recruitment (according to WHO charts) [13,14] were very similar (*p* = 0.436 for weight, *p* = 0.638 for length/height and *p* = 0.975 for head circumference, using *t*-test) (Figure 1a–c).

Significant individual positive correlations between percentiles of length and head circumference at birth and of height and head circumference at follow-up were detected, respectively with a moderate and a weak correlation Figure 2, with a *p*-value of 0.004 for length/height and of 0.046 for head circumference.

An increase of one percentile of length at birth determined an increase of 0.352 percentiles of height at follow-up, while an increase of one percentile of head circumference at birth determined an increase of 0.214 percentiles of head circumference at follow-up.

Children who were SGA at birth showed a significant increased risk of unsatisfactory growth and need to auxological referral (χ^2^ = 4.595; *p* = 0.032), while subjects who were identified as IUGR during intrauterine life (being or not SGA at birth) did not (*p* = 0.676).

Regarding morbidities and feeding characteristics, hospitalization at birth had an average duration of 25.68 ± 20.45 days and 28 infants (35%) were hospitalized again during the first year of life. Thirty-three children (41.25%) are in clinical follow-up because of minor problems not affecting feeding nor growth and 6 subjects (7.5%) have food allergies. Vaccination rate was high in the evaluated sample: 78 children (97.5%) were regularly vaccinated, 1 child was exempt due to severe immunodeficiency and only 1 had a delayed vaccination schedule. No socioeconomic problems were present in children’s families.

In the first six months of life 53.8% of infants were exclusively or predominantly breastfed, 46.2% were exclusively formula-fed, using standard formulas for preterm and then for term infants in the great majority of cases. Average weaning time was 6.00±1.09 months. No infant followed special diets and 19 children (23.75%) were given food supplements (i.e., vitamins, iron) at the time of follow-up.

## 4. Discussion

The main findings of the present study were first, that the use of INTERGROWTH-21^ST^ size at birth Standards [9,10] and of preterm postnatal standards [11] thereafter, significantly reduces the diagnosis of short term EUGR (*p* < 0.0001) when compared to routinely used size at birth charts, among whom Fenton charts [12] were chosen because they are widely employed in clinical practice. Second, INTERGROWTH-21^ST^ size at birth standards [9,10] harmonize well with WHO child growth standards [13,14], at least until preschool age.

Finally, being born SGA on, INTERGROWTH-21^ST^ size at birth standards [9,10] is predictive of an unfavorable growth outcome at preschool age.

This study was conducted on a population of preterm infants admitted to a special care neonatal unit, without chronic or major morbidities. Most the infants were moderate and late preterm, who were reported in literature to be a population at risk of growth impairment and adverse outcome later in life, even if not as much as the severely preterm infants [6,8]. 

For such a vulnerable group of infants, the growth during hospitalization and the prevalence of growth failure at discharge were evaluated. A longitudinal approach is advisable in order to assess the EUGR, which can be calculated in different ways, with the percentile loss between birth and discharge or 40 weeks postmenstrual age, whichever comes first, being the most credited. In our study, the employment of INTERGROWTH-21^ST^ Preterm Postnatal Growth Standards [11] significantly reduced the diagnosis of short term EUGR compared to neonatal cross-sectional charts [12] widely used in literature. [19] Infants could therefore be less forced to rapid weight gain, as their growth trajectory appeared more satisfactory.

The small percentile loss experienced at discharge was recuperated over time and the distribution of percentiles at recruitment (on WHO standards) was very close to percentiles at birth in our cohort. For this cohort of moderate and late preterm infants, who did not develop relevant morbidities over the follow-up period, WHO charts [13,14] represented the ideal prosecution of INTERGROWTH-21^ST^ standards [9,10,11], in order to evaluate growth from birth to the preschool age.

Finally, a focus was put on SGA infants, who are known to be at increased risk of growth faltering and developing complications. [3,4,20,21,22] It is of clinical relevance that a growth chart helps the clinician in identifying infants at risk, enabling a strict follow-up of more vulnerable subgroups. In this study, being SGA at birth was significantly associated with unsatisfactory growth at follow-up, i.e., statural height lower than third percentile on WHO standards and/or height growth percentile falling outside the parents’ target and/or reduced growth velocity for sex and age calculated longitudinally with at least two measurements during follow-up, thus suggesting that charts were able to early select a high-risk subgroup. A history of intrauterine growth restriction alone was not significantly associated with later growth impairment, but this result should be taken with caution and possibly represents a limit of the study, as obstetric charts were retrospectively analyzed and prenatal ultrasounds not always available.

Recruited children did not show significant morbidities during follow-up, average weaning age was in line with guidelines and there was high adherence to vaccination schedule.

The main strengths of the study were the relatively large sample if considered that reliable and accurate measurements were taken for all subjects at least at three time points, i.e., birth, term corrected age and preschool age; the study has a major novelty that relies in its extension not only up to term corrected age, but to preschool age, and validates the use of INTERGROWTH-21ST and WHO Child growth standards as a consistent tool to monitor growth in the first years of life for preterm infants; finally, the gestational age at birth range between 30 and 36 weeks allows a good focus on moderately and late preterm infants, thus excluding confounding factors such as extremely low gestational age with its known burden of comorbidities and nutritional challenges.

Limitations of the study are the retrospective collection of obstetric antenatal data, the use at follow-up visits of questionnaires, which may be affected by recall bias and the fact that approximately one third of the intended preterm infant sample could not be included, for parental refusal or invalid phone numbers.

## 5. Conclusions

The employment of INTERGROWTH-21st charts [11] during the follow-up significantly reduces the diagnosis of short term EUGR, leading to a less aggressive nutritional approach [19]; moderate and late preterms without short-term EUGR based on specific preterm postnatal charts show adequate growth achievements at preschool age.

The implementation of a package of standards at birth, preterm postnatal growth standards [9,10,11] and WHO child growth standards [13,14] proved to be consistent and effective in monitoring growth in a moderate and late preterm infant cohort without chronic or major morbidities.

Size at birth neonatal standards [9,10] identified, in those born below 10th percentile for weight, a subgroup with significantly higher risk of slower catch-up growth during childhood and need of auxological referral, thus well predicting growth impairment at pre-school age.

The present study was conducted on a moderate and late preterm infants cohort: future studies focused on very preterm neonates may shed light on the growth monitoring of this high risk population, even if those infants are more difficult to be followed-up on INTERGROWTH-21^ST^ existent charts, that are more reliable after 32 weeks of gestational age. Those infants are also more complicated to study because of frequent comorbidities and still debated nutritional strategies.

## Figures and Tables

**Figure 1 nutrients-12-01941-f001:**
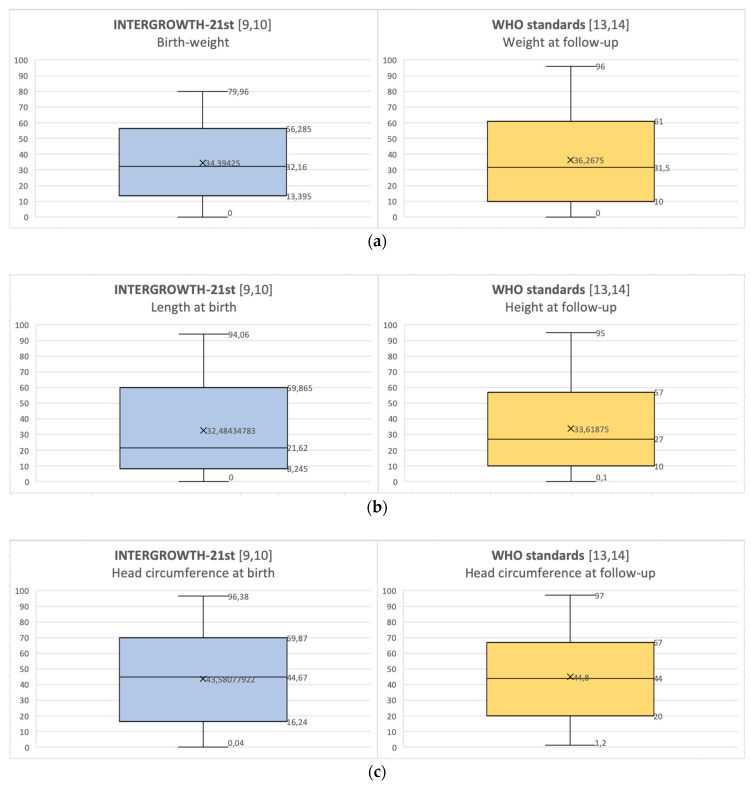
(**a**) Distributions of weight percentiles at birth and at follow-up; (**b**) distributions of length/height percentiles at birth and at follow-up; (**c**) distributions of head circumference percentiles at birth and at follow-up.

**Figure 2 nutrients-12-01941-f002:**
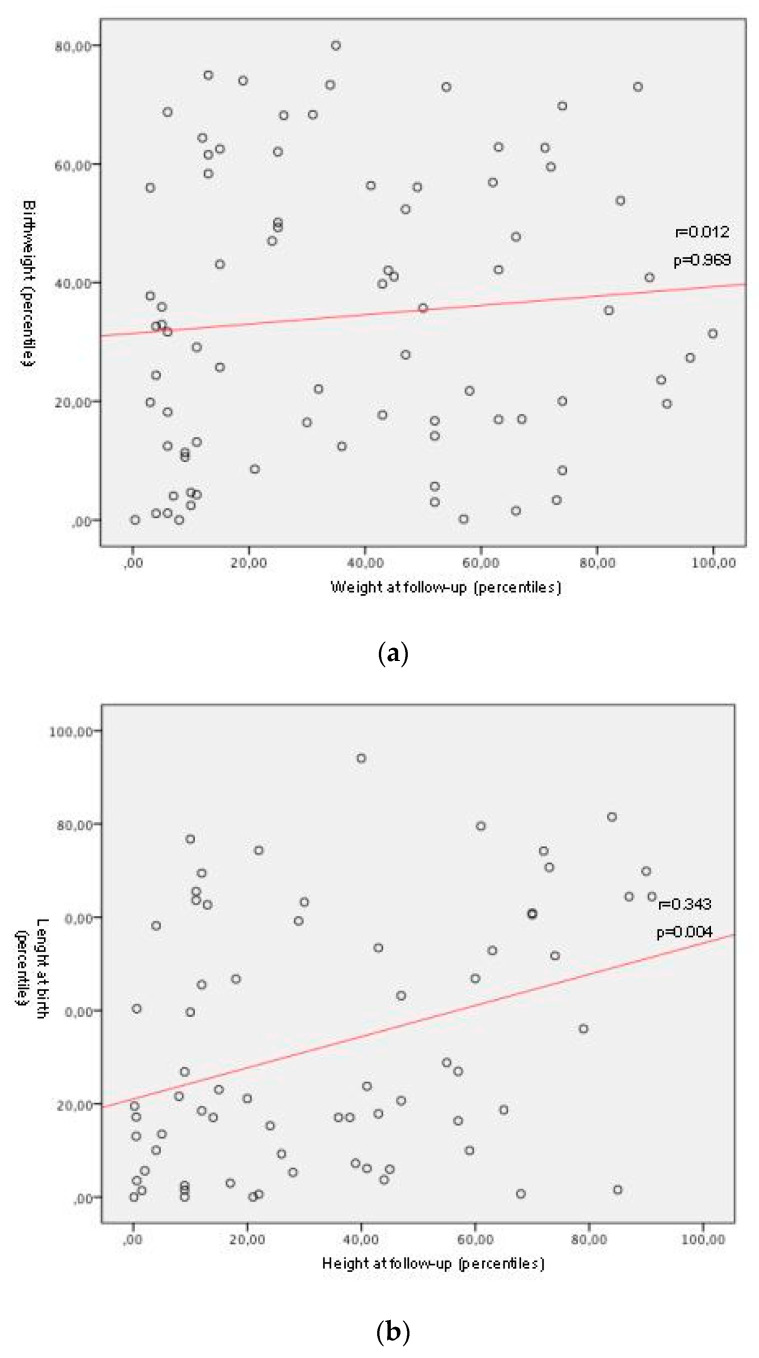
Correlation between percentiles at birth and at follow-up for weight, length or height, head circumference.

**Table 1 nutrients-12-01941-t001:** Baseline characteristics.

Baseline Characteristics of Study Population (*n* = 80)	
**Male gender (*n*, %)**	46 (57.5%)
Cesarean section (*n*, %)	37 (67.2%)
Maternal smoking in pregnancy	1 (1.25%)
Hypertension in pregnancy	8 (10%)
Gestational diabetes	10 (12.5%)
Average age at follow-up ±SD	4.21±0.28 years
Twins (*n*,%)	50 (62.5%)
Average gestational age at birth ± SD	33.3 ± 2.2 weeks, range 30–36 weeks
Average birthweight ± SD	1835 ± 486 g
Average weight z-score at birth	−0.61 ± 0,97
Average percentile at birth [9,10] ± SD	Weight: 34.39 ± 23.88
	Length: 32.48 ± 26.91
Average percentile at birth [9,10] ± SD SGA at birth (n, %) IUGR history (n, %)	Head circumference: 43.58 ± 29.82
	15 (18.75%)
	13 (16.25%)
Length of stay, average ± SD	25.68 ± 20.45 days
Average percentile at follow-up [13,14] ± SD	Weight: 36.73 ± 28.10
Length of stay, average ± SD	Height: 33.62 ± 27.69
Average percentile at follow-up [13,14] ± SD Average weight z-score at follow-up Average height z-score at follow-up	Head circumference: 44.80 ± 27.21
	−0.42 ±1.04
	−0.63 ± 1.06

SGA: small for gestational age; IUGR: intrauterine growth restriction.

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
