# Peer review of "Growth Assessment in Preterm Children from Birth to Preschool Age"

_nutrients, 2020, doi:10.3390/nu12071941_

Round 1
Reviewer 1 Report
Summary: Ceratto and collaborators evaluate growth of 80 children born preterm using new INTERGROWTH-21st and WHO Child Growth Standards compared with size at birth-growth charts. The aim was to evaluate in-hospital growth, growth failure at discharge, long-term growth until preschool age, and assess predictivity of neonatal growth on later childhood growth. Their main finding was that using new growth standards reduced the diagnosis of extrauterine growth restriction.
Broad comments: This is an important topic. However, the manuscript needs much rewriting and the language needs editing. I made a few examples regarding this, there are more to be corrected. I found it difficult to quickly find the main findings, especially in the Abstract.
Specific comments to Authors:
- The topic is relevant and the descriptive headline goes well with it.
- All abbrevations should be defined in parentheses the first time they appear in the abstract (e.g. INTERGROWTH-21st, line 19; EUGR line 21, WHO line 23), key words (SGA line 29), main text (INTERGROWTH-21st, WHO, SGA) and be used consistently thereafter. Also, I recommend defining abbreviations as footnotes in tables (SGA, IUGR).
- The Abstract should be more informative. At least the method needs to be described, i.e. comparing new vs. old type growth curves. Further, some information on the participants should be provided. I suggest at least including data on gestational age (mean, SD) and sex (n, % for either girls or boys). Further, the results could be summarized briefly and the conclusion needs clarification.
- Is there any reference for the older, size at birth charts (line 54)?
- What was the hypothesis? I recommend inserting the hypothesis, together with the aim of the study, at the end of the Introduction.
- On line 65, regarding exclusion criteria could be rewritten as “Syndromes, conditions or diseases known to…”
- I assume that written consent was obtained from the parent or guardian, not the children. This should be stated.
- Lines 71-72 need rewriting, for example “All participants underwent a physical examination. Parents completed a questionnaire enquiring data on the child’s health, feeding practices, acquired vaccinations and reaching the developmental milestones.”
- How was falling outside the parents’ target calculated? This should be mentioned in the Methods (line 80).
- Participant selection is usually described in the Methods, not in the Results (lines 85-88), finishing with for example “Finally, 80 children born preterm were included in the study”. This should be changed.
- In Table 1 I recommend including total participant number, birth weight, duration of stay at hospital, breastfeeding and data on common maternal pregnancy related conditions (pre-eclampsia, gestational diabetes, smoking during pregnancy), if available. Also family socioeconomic status (income, parental education) would be of interest.
- On lines 109-111, the Authors describe correlations between size at birth and measurements at the 4-year visit. For the common reader it would perhaps be easier to understand how an increase of one percentile of length at birth turns into XXcm at the follow-up visit. Instead of simply stating 0.352 percentiles of height, could you also give the corresponding value in cm?
- I recommend starting the Discussion with a brief summary of the key findings.
- What were the strengths and limitations of this study? Many other factors than preterm birth have effects on the growth of children. For example family socioeconomic status and parental BMI are known to effect family diet. Were these children firstborn? Firstborn children tend to have smaller birth weights than the following siblings. How did the Authors account for confounders? Which ones?
Author Response
Thank you for your positive evaluation and for you thoughtful comments that have been all addressed.
Re your specific comments, please find our answers. Substatial changes have been made to the manuscript following your suggestions:
1. The topic is relevant and the descriptive headline goes well with it.
Thank you for the positive comment
2. All abbrevations should be defined in parentheses the first time they appear in the abstract (e.g. INTERGROWTH-21st, line 19; EUGR line 21, WHO line 23), key words (SGA line 29), main text (INTERGROWTH-21st, WHO, SGA) and be used consistently thereafter. Also, I recommend defining abbreviations as footnotes in tables (SGA, IUGR).
All abbreviations have been defined in parentheses the first time they appear in the abstract and in the text (revised manuscript lines 20, 49, 57, 62. As requested, abbreviations have been defined as footnotes in tables (line 131). Moreover, as requested by the journal, the abbreviations list is available at the end of the manuscript.
3. The Abstract should be more informative. At least the method needs to be described, i.e. comparing new vs. old type growth curves. Further, some information on the participants should be provided. I suggest at least including data on gestational age (mean, SD) and sex (n, % for either girls or boys). Further, the results could be summarized briefly and the conclusion needs clarification.
The abstract has been revised accordingly to your comments. The comparison between new and old type of growth curves has been detailed (lines 22-24) and some information on the participants is now provided (lines 18-19). Results have been summarized and conclusions rewritten (lines 21 - 31)
4. Is there any reference for the older, size at birth charts (line 54)?
Reference 14 has been added, line 68 of the revised manuscript
5. What was the hypothesis? I recommend inserting the hypothesis, together with the aim of the study, at the end of the Introduction.
Thank you for suggesting this, hypothesis and aim of the study have been detailed at the end of the introduction, lines 56-64
6. On line 65, regarding exclusion criteria could be rewritten as “Syndromes, conditions or diseases known to…”
Exclusion criteria have been rewritten as suggested, lines 79-80 of the revised paper
7. I assume that written consent was obtained from the parent or guardian, not the children. This should be stated.
Consent was obtained from the parent or guardian as you assume and this has been specified, lines 120-121
8. Lines 71-72 need rewriting, for example “All participants underwent a physical examination. Parents completed a questionnaire enquiring data on the child’s health, feeding practices, acquired vaccinations and reaching the developmental milestones.”
The sentence describing follow-up was changes according to your input, lines 97-99
9. How was falling outside the parents’ target calculated? This should be mentioned in the Methods (line 80).
Parents’ target calculation is now included in the methods and referenced (lines 116-117)
10.Participant selection is usually described in the Methods, not in the Results (lines 85-88), finishing with for example “Finally, 80 children born preterm were included in the study”. This should be changed.
Participant selection has been moved to Methods section, lines 83-86
11. In Table 1 I recommend including total participant number, birth weight, duration of stay at hospital, breastfeeding and data on common maternal pregnancy related conditions (pre-eclampsia, gestational diabetes, smoking during pregnancy), if available. Also family socioeconomic status (income, parental education) would be of interest.
Table 1 has been enriched with the following data as requested: total participant number (headline ), birth weight , duration of stay at hospital, data on common maternal pregnancy related conditions (Hypertension in pregnancy, gestational diabetes, smoking during pregnancy).
Socioeconomic status and breastfeeding have been added to the text, lines 165-166 and 167-169.
12. On lines 109-111, the Authors describe correlations between size at birth and measurements at the 4-year visit. For the common reader it would perhaps be easier to understand how an increase of one percentile of length at birth turns into XXcm at the follow-up visit. Instead of simply stating 0.352 percentiles of height, could you also give the corresponding value in cm?
The entity of the difference in millimetres depends on GA at birth and age at follow up, as well as on the specific percentile, as central centiles are closer than extreme centiles. It cannot be quantified except individually.
13. I recommend starting the Discussion with a brief summary of the key findings.
Your suggestion has been followed and a new paragraph summarizing the key finding has been inserted at the beginning of the discussion, lines 173-180
14. What were the strengths and limitations of this study? Many other factors than preterm birth have effects on the growth of children. For example family socioeconomic status and parental BMI are known to effect family diet. Were these children firstborn? Firstborn children tend to have smaller birth weights than the following siblings. How did the Authors account for confounders? Which ones?
Re firstborn/following siblings, nor INTERGROWTH-21ST nor WHO charts are specific for parity, so this was not considered.
Re strengths and limitations of the study, those are now added in detail in the discussion section, lines 210-219
Reviewer 2 Report
Although INTERGROWTH-21st standards overlap the World Health Organization Child Growth Standards (conceived for term-born children) at about 6 months’ corrected age, an ideal standard (prescriptive) constructed from a large long-term follow-up cohort of “healthy” infants, including neonates from the threshold of viability to term gestational age at birth, allowing their use through all life period is lacking (Perumal 2015, Pereira-da-Silva 2019). Therefore, it is important to investigate which charts can mitigate this lack and can be reasonably used to assess long-term growth of children born preterm.
In this context, this study had the interesting objective of evaluating if the WHO Child Growth Standards could assess accurately the current growth of school age children born preterm, evaluating retrospectively the consistency of the current growth with growth of the same sample assessed at birth and at term corrected age using the INTERGROWTH-21st standards.
This study has several flaws that deserve critics and need clarification of some aspects:
- For this study, a sample of 237 preterm neonates born during a period of 13 months was eligible, but that results may be not representative of the sample that was intended to be studied, because only one third (33.7%) has been analyzed. This should be acknowledged as a major limitation of the study.
- To compare assessments using standards obtained from different populations, the z-scores seems to be more precise that percentiles (Ohuma 2019) and both INTERGROWTH-21st and WHO Child Growth Standards have z-scores available.
- The study design should be specified in Methods.
- The stated (line 54) “widely used size-at birth charts” should be appropriately referenced.
- Exclusion criteria should be better defined, by specifying the main conditions (line 65) that have been considered for exclusion.
- The manuscript suggests that the study was based on three points of assessment for anthropometry: at birth, at term corrected age, and 4-5 years of age. However, other points of assessment for follow-up are stated in lines 76-77 and 81. Please clarify who have measured the infants, if measurements were undertaken in all 80 children analyzed, and how reliable these measurements were to be considered.
- Also, some results from clinical follow-up is presented (lines 116-125) and it should be specified in Methods, the source of these data and if they were reliable.
- Criteria for defining IUGR presented in Results must be specified in Methods, supported with appropriate references.
- It is stated (line 113) that “SGA at birth showed a significant increased risk of unsatisfactory growth”. Which measurements have been used to assess growth (bodyweight, stature, weight-to-stature ratio, head circumference)?
- Where (lines 21, 133, 157) it is stated “EUGR” it should be “short-term EUGR”
- The authors suggest the development of future studies focused on very preterm infants. However, to maintain this suggestion it should be clarified that INTERGROWTH-21st standards cannot be used, because only 28 infants born at 33 weeks gestation or earlier contributed data to these standards, and they are only reliable for monitoring postnatal growth only from 32 weeks postmenstrual age in infants born at more than 27 weeks of gestation (Pereira-da-Silva 2019).
References
- Ohuma EO, Altman DG; International Fetal and Newborn Growth Consortium for the 21st Century (INTERGROWTH-21st Project). Statistical methodology for constructing gestational age-related charts using cross-sectional and longitudinal data: The INTERGROWTH-21st project as a case study. Stat Med. 2019;38(19):3507‐3526.
- Pereira-da-Silva L, Virella D, Fusch C. Nutritional assessment in preterm infants: a practical approach in the NICU. Nutrients. 2019;11(9):1999.
- Perumal N, Gaffey MF, Bassani DG, Roth DE. WHO Child Growth Standards are often incorrectly applied to children born preterm in epidemiologic research. J Nutr. 2015;145(11):2429‐2439.
Author Response
Thank you for your positive evaluation and for you thoughtful comments that have been all addressed.
Re your specific comments, please find our answers. Substatial changes have been made to the manuscript following your suggestions:
- For this study, a sample of 237 preterm neonates born during a period of 13 months was eligible, but that results may be not representative of the sample that was intended to be studied, because only one third (33.7%) has been analyzed. This should be acknowledged as a major limitation of the study.
ANSWER
Thank you very much for noticing this, that is a mistake in presenting the data. 237 were all infants admitted to the Unit, but of those about half were term infants and only 124 were preterm and without major morbidities.
Of the 124 preterms, 80 accepted and were included, making the studi sample of 65% of the intended one. Text has been corrected, lines 83-86
- To compare assessments using standards obtained from different populations, the z-scores seems to be more precise that percentiles (Ohuma 2019) and both INTERGROWTH-21st and WHO Child Growth Standards have z-scores available.
ANSWER
z-scores values for birweight, weight at follow-up and height at follow-up have been calculated and incorporated in Table 1
- The study design should be specified in Methods.
ANSWER
Study design has been added in Methods section, line 76
- The stated (line 54) “widely used size-at birth charts” should be appropriately referenced.
ANSWER
Reference has been added, line 68 of the revised paper
- Exclusion criteria should be better defined, by specifying the main conditions (line 65) that have been considered for exclusion.
ANSWER
Exclusion criteria have been revised as suggested and examples of specific conditions have been added, lines 79-80
- The manuscript suggests that the study was based on three points of assessment for anthropometry: at birth, at term corrected age, and 4-5 years of age. However, other points of assessment for follow-up are stated in lines 76-77 and 81. Please clarify who have measured the infants, if measurements were undertaken in all 80 children analyzed, and how reliable these measurements were to be considered.
ANSWER
The study was based on three points of assessment for anthropometry, which as you state were birth, term corrected age and 4-5 years of age.
Clarifications about other points of assessment have been added in the methods, lines 110-112.
Details about who have measured the infants, if measurements were undertaken in all 80 children analyzed, and how reliable these measurements were to be considered have been added in methods section, lines 92- 96, and in the discussion, lines 210 – 212..
- Also, some results from clinical follow-up is presented (lines 116-125) and it should be specified in Methods, the source of these data and if they were reliable.
ANSWER
Sources of all data have been listed in the Methods section, lines 87 for antenatal data, lines 90 – 91 for neonatal data, lines 97-99 for follow-up data. Reliability id discussed as suggested, lines 206-207 and 215-216.
- Criteria for defining IUGR presented in Results must be specified in Methods, supported with appropriate references.
ANSWER
Thank you for your comment, definition and reference have been added in the methods section, lines 88-89.
- It is stated (line 113) that “SGA at birth showed a significant increased risk of unsatisfactory growth”. Which measurements have been used to assess growth (bodyweight, stature, weight-to-stature ratio, head circumference)?
ANSWER
At birth and follow-up, body weight, length or height and head circumference were considered. SGA was defined as with birth weight below 10th percentile
- Where (lines 21, 133, 157) it is stated “EUGR” it should be “short-term EUGR”
ANSWER
Agree, text has been changed . In the revised paper, lines are 22, 175, 191, 223
- The authors suggest the development of future studies focused on very preterm infants. However, to maintain this suggestion it should be clarified that INTERGROWTH-21st standards cannot be used, because only 28 infants born at 33 weeks gestation or earlier contributed data to these standards, and they are only reliable for monitoring postnatal growth only from 32 weeks postmenstrual age in infants born at more than 27 weeks of gestation (Pereira-da-Silva 2019).
ANSWER
A statement has been added at the end of the conclusions, lines 236-238
Reviewer 3 Report
The manuscript “Growth assessment in preterm children from birth to preschool age” has been carefully reviewed. There are some considerable concerns/amendments that need to be carried out. The reasons have been explained below why I reach this conclusion. Please consider carefully all of the suggestions.

Author Response
Thank you for your comments, that have been accurately addressed.
Re your specific comments, please find our answers. Substatial changes have been made to the manuscript following your suggestions:
- Data are new and from original research on a cohort of preterm infants admitted to a Special Care Unit. A paragraph has been added to the discussion to discuss strength and weaknesses of the study, lines 210-216
- Following your suggestion, a paragraph has been written to clarify hypotheses and aims of the study and how this will contribute to existing knowledge, lines 56-71 of the revised manuscript
- The Materials and Methods have been thoroughly revised and almost entirely rewritten to be clearer and more informative to the reader, lines 73 – 123. In particular, details have been added regarding the exclusion criteria, the data sources , the accuracy of anthropometry, the number of measurements, parental target ecc.
4 - 5. Results and table 1 have been clarified and enriched in order to be more informative and precise
- Growth rate can not be calculated reliably between two distant point, i.e. birth and 4-5 years of age. Average absolute measurements and z-scores were used for the comparison between the two groups. Kind of feeding in the first six months of life has been indicated but it is unlikely to be a major determinant of body size at 4-5 years of age
- Discussion has been revised with a brief summary of key findings and with a paragraph about strengths and limitations of the study.
- No results from previous studies are even mentioned in the paper, nor considered, because the study is based on an ad hoc selected infant cohort. The study has a major novelty that relies in its extension not only up to term corrected age but to preschool age, and validates the use of INTERGROWTH-21ST and WHO Child growth standards as a consistent tool to monitor growth in the first years of life for preterm infants. Conclusions have been revised and clarified.
Round 2
Reviewer 2 Report
Most of questions have been responded, however few have not been addressed:
- Although the sample dimension intended to be studied has been corrected and is lower than formerly stated, around one third 35.5% of infants were not included and may still affect the representativeness of the results. As I have suggested, this must be acknowledged as a limitation of the study.
- Line 80: please explain the meaning of the abbreviation “ecc”
- Lines 203-204: The anthropometric measurements at follow-up that support the statement “associated with increased risk of unsatisfactory growth” should be specified.in the text
- Line 58: I suggest stating “calculated using inappropriate charts” instead of “calculated on inappropriate charts”
- Line 216: correct typo “excluding”
The manuscript needs an English editing by an English native speaker or by an English editing service.
Author Response
Thank you for your additional comments, that have been all addressed.
Re your specific comments, please find below our answers.
- Although the sample dimension intended to be studied has been corrected and is lower than formerly stated, around one third 35.5% of infants were not included and may still affect the representativeness of the results. As I have suggested, this must be acknowledged as a limitation of the study.
Included, lines 239-241
- Line 80: please explain the meaning of the abbreviation “ecc”
Typing error, amended, line 80
- Lines 203-204: The anthropometric measurements at follow-up that support the statement “associated with increased risk of unsatisfactory growth” should be specified.in the text
Added, lines 221-223
- Line 58: I suggest stating “calculated using inappropriate charts” instead of “calculated on inappropriate charts”
Amended, line 58
- Line 216: correct typo “excluding”
Amended, line 236
- The manuscript needs an English editing by an English native speaker or by an English editing service.
We shall ask English editing service from the journal, upon manuscript acceptance for publication
Reviewer 3 Report
The revised manuscript “Growth assessment in preterm children from birth to preschool age” has been carefully reviewed. Thank you for your response to my suggestion, but your revised manuscript were not fully considered majority of considerable concerns/amendments. Please consider carefully all of previous comment and present suggestions. And please explain your modifications paired with each comment.
- The overall content still tends to be unorganized and distracting.
- The explanation of statistical method has not revised, which is too short and weak. For example, line 152-154: There is no description what kind of statistical analysis you used.
- I wonder if there is any specific reason to make table 2. I think that graphics are more proper and effective to show the correlation.
- The discussion is still very weak. In results, you mentioned that you didn’t include socio-economic information of study population, which might influence on the growth rate of infants. However, this was not considered in discussion.
Previous comment
- There are not enough data for the paper or to support your subject and also data are not innovative.
- Regarding to the originality of the manuscript in the end of Introduction, it should be indicated clearly how this review contributes to the existing knowledge. Please also identify and describe the originality of the study described in this investigation and how it will likely to contribute to the existing knowledge.
- The Materials and Methods are not explained adequately to achieve the proposed aims.
The adverse outcome of SGA infants are mentioned through entire article, but there is no definition what the adverse outcome of SGA is. The explanation of statistical method is too short and weak.
- The results are not well-organized, which makes very difficult to find out what your study wants to investigate. The contents should be summarized as table or figure to make your results clear and precise.
- The form and contents of tables and figures are not appropriate, nor strong enough to show your results.
- Considering the subject, it needs to present the differences of grow rate according to SGA /non-SGA. And also the kind of feeding is mentioned in results. It also needs the data of the differences of grow rate according to the kinds of feeding.
- The discussion is very weak. It dose not include enough reviews and limitations of your study to support its subject.
- The conclusion is not suitable since there is no novelty from your study. It is just summarized the results of previous studies.
Author Response
Thank you for your comments, please find below our answers. Changes have been made to the manuscript following your suggestions:
- Results have been divided into paragraphs, with a partition between birth and in-hospital growth and follow-up results, lines 132, 145
- Statistical analyses used are now detailed in the methods section, lines 123-124
- Following your suggestion, we removed table 2 and replaced with correspondent Figure 2, with more proper graphics (lines 165 - 172)
- Results may have been not clear and lines 182 - 183 have been slightly changed. We included those information, however “no socio-economic problems were present in children’s families”, so there is no effect of this variable on our population subjects.